# Effects of a High-Concentrate Diet on the Blood Parameters and Liver Transcriptome of Goats

**DOI:** 10.3390/ani13091559

**Published:** 2023-05-06

**Authors:** Yusu Wang, Qiong Li, Lizhi Wang, Yuehui Liu, Tianhai Yan

**Affiliations:** 1Institute of Animal Nutrition, Sichuan Agricultural University, Chengdu 611130, China; 2Livestock Production Sciences Branch, Agri-Food and Biosciences Institute, Large Park, Hillsborough BT26 6DR, UK

**Keywords:** goats, high-concentrate diet, liver, LPS, glycolipid metabolism, transcriptome

## Abstract

**Simple Summary:**

In order to increase the growth performance and make up for the lack of high-quality roughage, farmers are accustomed to adopting high-concentration diets during the fattening period of ruminants. The long-term consumption of high-concentration diets induces a series of nutritional metabolic diseases. In production practice, the major mutton sheep fattening region (e.g., Jiangsu, China) generally uses diets with a concentrate ratio of up to 80% to 90% during the fattening period, and reports of cases of nutritional metabolic diseases are also rare. This seems to indicate that sheep and goats are more adaptable to high-concentration diets than cattle. Our work shows that high-concentrate diets can significantly promote the digestion of nutrients; the liver enhances the adaptability of goats to high-concentration diets by regulating the expression of genes involved in nutrient metabolism and toxin clearance. Therefore, the liver plays a vital role in the adaptation of ruminants to high-concentration diets. The results here can lay the foundation for the rational application of high-concentration diets in production.

**Abstract:**

The objective of this study was to determine the effect of high-concentrate diets on the blood parameters and liver transcriptome of goats. Eighteen goats were allocated into three dietary treatments: the high level of concentrate (HC) group, the medium level of concentrate (MC) group, and the low level of concentrate (LC) group. The blood parameters and pathological damage of the gastrointestinal tract and liver tissues were measured. In hepatic portal vein blood, HC showed higher LPS, VFAs, and LA; in jugular vein blood, no significant differences in LPS, VFAs, and LA were recorded among groups (*p* > 0.05). Compared to the LC and MC groups, the HC group showed significantly increased interleukin (IL)-1β, IL-10, TNF-α, and diamine oxidase in jugular vein blood (*p* < 0.05). Liver transcriptome analysis discovered a total of 1269 differentially expressed genes (DEGs) among the three groups and most of them came from the HC vs. LC group. There were 333 DEGs up-regulated and 608 down-regulated in the HC group compared to the LC group. The gene ontology enrichment analysis showed that these DEGs were mainly focused on the regulation of triacylglycerol catabolism, lipoprotein particle remodeling, and cholesterol transport. The Kyoto Encyclopedia of Genes and Genomes pathway analysis revealed that the liver of the HC group enhanced the metabolism of nutrients such as VFAs through the activation of AMPK and other signaling pathways and enhanced the clearance and detoxification of LPS by activating the toll-like receptor signaling pathway. A high-concentrate diet (HCD) can significantly promote the digestion of nutrients; the liver enhances the adaptability of goats to an HCD by regulating the expression of genes involved in nutrient metabolism and toxin clearance.

## 1. Introduction

In order to increase the growth performance and make up for the lack of high-quality roughage, farmers are accustomed to adopting high-concentration diets (HCDs) during the fattening period of ruminants [1,2,3,4]. The long-term consumption of an HCD will cause rumen microflora disorders and abnormal rumen fermentation and body metabolism, thus inducing a series of nutritional metabolic diseases (NMDs), such as subacute ruminal acidosis (SARA), laminitis, and liver abscess [5], among others. However, the scientific reports on NMDs induced by HCD have been mostly focused on cattle, and few on goats and sheep. The results of evolutionary adaptation research suggest that the rumen of small ruminants has a higher buffering capacity compared to cattle due to the differences in digestive tract anatomy [6]. In production practice, the major mutton sheep fattening region (e.g., Jiangsu, China) generally uses diets with a concentrate ratio of up to 80% to 90% during the fattening period, and reports of cases of NMDs are also rare. These seem to indicate that sheep and goats are more adaptable to HCDs than cattle, but further research is needed to confirm this hypothesis. The consumption of HCD can lead to a substantial increase in the production of nutritional metabolites such as volatile fatty acids (VFAs) and lactic acid (LA), as well as toxic substances in the rumen such as ammonia and lipopolysaccharides (LPS) [7,8]. After these substances are absorbed into the body, they are first delivered to the liver through the hepatic portal vein, where they are metabolized or detoxified, and then transported into peripheral tissues [9]. Therefore, the liver plays a vital role in the adaptation of ruminants to HCDs. The response of the forestomach, small intestine, and hindgut to HCDs has been widely studied [10,11,12], but so far there is little information on the response of the liver’s function and tissue morphology to an HCD. Based on the above questions, this experiment hypothesized that feeding goats a high-concentrate diet would not be harmful to the goats, where the liver played a role in metabolizing toxins. This experiment, therefore, was conducted to study the effects of an HCD on goat blood, rumen, and intestinal tissue morphology, and transcriptome sequencing technology was used to explore the functional response of goats’ liver to an HCD. The results can lay the foundation for the rational application of HCDs in production.

## 2. Materials and Methods

The research procedure used in the current study was approved by the animal policy and welfare committee of the agricultural research organization of Sichuan Province, China, and is in agreement with the rules of the Animal Care and Ethical Committee of the Sichuan Agricultural University (Ethics Approval Code: SCAUAC201408-3).

### 2.1. Experimental Animals, Design, and Diet

This study was conducted at the experimental farm of the Animal Nutrition Institute, Sichuan Agricultural University (Ya’an, China). Eighteen Jianzhou big ear goats (4 months of age, 9 males and 9 females, BW = 20.26 ± 2.63 kg) were randomly allocated into 3 dietary treatments (6 replicates/treatment) as follows: high level of concentrate (HC) group, concentrate:roughage (C:R) = 80:20; medium level of concentrate (MC) group, C:R = 50:50; and low level of concentrate (LC) group, C:R = 25:75. The goats in each treatment group were divided equally between male and female. The experimental diet was formulated according to the Chinese Feeding Standard of Meat-producing Sheep and Goats (NY/T816-2004), and the ingredients and chemical composition are presented in Table 1. In order to avoid the interference of mycotoxins, especially aflatoxins (AFB1), with the experimental results, the diets were formulated with the effective control of aflatoxin content in the selection of raw materials [13] and with a focus on the quality and suitability of the hay [14]. The final preparation was TMR fed to goats [15]. All goats were housed individually in metabolic crates equipped with feeders and water buckets, and experimental diets (total mixed ration) were fed to goats for ad libitum intake in two equal amounts every day roughly at 9:00 a.m. and 5:00 p.m. Fresh water was available freely at all times. After a 14 d adjustment period, a formal trial was carried out for 67 days, with a 60 d feeding trial (FT), a 6 d digestion trial, and sampling on the second day of the digestion trial.

### 2.2. Sample Collection

After finishing the feeding adaptation period, the total feces produced by each goat from the manure collection board were collected daily for 6 successive days using collection bags. All feces (antisepticised with 10% formaldehyde, *w*/*w* of 3% of feces) were stored at −20 °C for later analysis. On the day after the digestion trial, 4 tubes of blood samples from the jugular vein (JV) were collected with a 5 mL vacuum blood collection vessel 2 h after morning feeding; two were collected in a non-anticoagulant tube for serum preparation and two were collected in a tube containing EDTA for plasma preparation. All the sera and plasma were stored at −20 °C for further measurement. After sampling the JV blood, the experimental goats were anesthetized with ether and the abdominal cavity was immediately opened. The hepatic portal vein (HPV) was isolated and the blood sample was collected (the processing was the same as that of the JV blood collection). Then, the goats were slaughtered by bleeding from the JV. The slaughter process was conducted according to the conventional procedures of Sichuan Agricultural University. The liver, rumen, jejunum, and colon were separated. A piece of liver (10 g) from the right lobe was harvested within 30 min after slaughter and quickly stored in liquid nitrogen for later transcriptome sequencing. A piece of liver tissue from the left lobe and rumen tissue from the ventral blind sac (2 cm × 2 cm) and a 4 cm-long looped section from mid-jejunum and mid-colon were harvested, respectively. All tissues (liver, rumen, jejunum, and colon) were rinsed with ice-cold and sterile phosphate buffer saline (PBS; pH 7.4) and immediately fixed in 4% paraformaldehyde (PFA) (Sigma, St. Louis, MO, USA) at room temperature for pathological section preparation.

### 2.3. Laboratory Analyses

All feed and fecal samples dried at 55 °C were measured for the contents of dry matter at 105 °C (DM), crude protein (CP, Kjeldahl method), and ether extract (EE, Soxhlet extraction method) according to the methods described by the AOAC [16]. The neutral detergent fiber (NDF) and acid detergent fiber (ADF) were determined using the method described by Van Soest et al. [17]. The gross energy (GE) was measured using an oxygen bomb calorimeter (PAR-6400 Calorimeter, Parr Instrument Company, Moline, IL, USA). The apparent digestibility of nutrients was calculated using the following formula:$$\mathrm{Apparent}\;\mathrm{digestibility}\;\mathrm{of}\;\mathrm{nutrient}\;\left(\%\right)=\frac{\mathrm{ingested}\;\mathrm{nutrient}-\mathrm{excreted}\;\mathrm{nutrient}\;\mathrm{in}\;\mathrm{feces}}{\mathrm{ingested}\;\mathrm{nutrient}}\times 100$$

The concentrations of alanine transferase (ALT), blood urea nitrogen (BUN), total protein (TP), albumin (ALB), triglyceride (TG), glucose (GLU), low-density lipoprotein (LDL-C), and high-density lipoprotein (HDL-C) in the sera of the JV and the concentrations of triglyceride (TG), glucose (GLU), and non-esterified fatty acid (NEFA) in the serum of the HPV were measured using a commercial kit (Zhongsheng Beikong Biotechnology Co., Ltd., Shanghai, China) by a full-automatic biochemical analyzer (Beckman AU5800, Brea, CA, USA). The concentrations of tumor necrosis factor-α (TNF-α), interleukin (IL)-6, IL-1β, IL-10, IL-4, and serum amyloid A protein (SAA) in the sera of the JV were determined by enzyme-linked immunosorbent assay (ELISA), and the kits were purchased from Shanghai Biyuntian Biotechnology Co., Ltd. (Shanghai, China). The concentrations of lipopolysaccharide-binding protein (LBP), haptoglobin (HP), and diamineoxidase (DAO) in the plasma of the JV were measured using commercial ELISA kits purchased from Shanghai Kangnang Biological Technology Co., Ltd. (Shanghai, China). The concentrations of LA and LPS in the JV and HPV plasma were determined using ELISA kits purchased from Jiangsu Sinitic Biological Technology Co., Ltd. (Xuyi, China). The concentrations of VFAs (acetic acid, propionic acid, butyric acid) in JV and HPV sera were determined by gas chromatography (Agilent GC6890, Santa Clara, CA, USA) with a column of 19,091 N-213. The measurement conditions were: inlet temperature 220 °C, carrier gas (N_2_) flow rate 2.0 mL/min, split ratio 40:1, injection volume 0.6 μL, and a 120 °C programmed ramp-up for 3 min, followed by a 10 °C/min ramp-up to 180 °C for 1 min. The hydrogen flame ionization detector (FID) was set at 250 °C with 40 mL/min hydrogen, 450 mL/min air, and 45 mL/min tailpipe gas. The tissue section was prepared by referring to a previously described method [18]. In brief, all tissue samples were separately cut into 2 cross sections 1 mm thick, washed, and then embedded in paraffin blocks. Subsequently, tissues were dehydrated and embedded in paraffin, then cut into 3 sections with a thickness of 5 mm, and then stained with Hematoxylin and Eosin for histological study under a light microscope. The histopathology observations, including the organizational structure and cell morphology of 3 slides for each sample were recorded by an optical binocular microscope (CX22, OLMPUS, Tokyo, Japan) fitted with a digital camera, which was equipped with the Image-Pro software (Leica, Inc., DM1000, Wetzlar, Germany). The pictures of normal tissues or areas with obvious lesions were taken and recorded using the microscopic imaging system. Tissue pathological damage of the rumen, liver, jejunum, and colon was scored from 1 to 3 referring to the previously described methods [19,20,21], respectively, with a score of 0 denoting no damage and a score of 3 denoting severe damage.

### 2.4. Transcriptome Analysis of the Liver

The liver tissue was ground into powder in a liquid nitrogen environment. The total RNA was extracted using Trizol reagent (Invitrogen Life Technologies, Carlsbad, CA, USA) and monitored on 1% agarose gels. After integrity, concentration, purity, degradation, and contamination tests, the satisfactory RNA samples (3 μg) were used as input material for cDNA library construction using the TruSeq RNA Sample Preparation Kit (Illumina, San Diego, CA, USA) following the manufacturer’s recommendations. The paired-end sequencing of the cDNA library on a HiSeq platform (Illumina) was performed by Shanghai Personal Biotech. Co. Ltd., Shanghai, China. The adaptor sequences and low-quality sequence reads (containing adapter reads, poly-N reads, and reads with a low base recognition rate) were removed from the raw data sets, and then clean data sets were obtained. At the same time, the Q20, Q30, and GC content of the clean sequence was calculated. The clean reads were mapped to the reference goat genome using Hisat2 (v2.0.5) and were de novo assembled to transcripts using Trinity software (v2.0.6). The assembled transcripts by Trinity were used as reference sequences and the clean sequence of each sample was mapped back onto the reference sequence using RSEM. The read count for each predicted gene was obtained according to the mapping results and the gene expression level was estimated by calculating the FPKM (fragments per kilobase of transcript sequence per million base pairs sequenced). Differential expression analysis among the 3 dietary treatments was performed using the DESeq R package, and the resulting *p* values were adjusted using Benjamini and Hochberg’s approach for controlling the false discovery rate (FDR) [22]. The genes with |log_2_ (fold change)| ≥ 1 and an adjusted *p*-value < 0.05 were considered as significantly differentially expressed genes (DEGs). GO (gene ontology) enrichment and KEGG (Kyoto Encyclopedia of Genes and Genomes) enrichment analysis of the DEGs was performed using DAVID and KOBAS, respectively.

### 2.5. Validating the Expression of DEGs by RT-qPCR

To validate the veracity and reliability of the transcriptome data, 8 DEGs were randomly selected for conducting real-time qPCR validation. The upstream and downstream primers of these genes (Appendix A) were designed using the NCBI’s Primer-BLAST and synthesized by Shenggong Bioengineering Co., Ltd. (Shanghai, China). Using β-actin as an internal reference gene, real-time qPCR was performed on a 7900HT system (Applied Biosystem, Foster City, CA, USA) using a qPCR Detection kit (SYBR Green, TIANGEN, Beijing, China). Data were processed using 7900HT Software (v.2.0.6) (Applied Biosystem) and the relative quantification of the gene expression differences was calculated with the formula 2^−ΔΔCt^.

### 2.6. Data Analysis

All data are presented as the mean ± SD. Statistical analyses were carried out using the SPSS statistical software (Ver. 20.0 for Windows; SPSS Inc., Chicago, IL, USA). The Shapiro–Wilk test and Levene’s test [23] were performed to test the data for normality and homoscedasticity, respectively. All data were subjected to a one-way analysis of variance (ANOVA) followed by Duncan’s multiple comparisons to determine significant differences among the treatments. The initial body weight was a covariate for statistical analysis. The 2^−ΔΔCt^ method was applied to analyze the real-time PCR data. Differences were considered significant at *p* < 0.05. The numbers of replicates used for statistics are noted in the Tables and Figures.

## 3. Results

### 3.1. The Digestibility of Nutrients

The digestibility of nutrients is shown in Table 2. The digestibility of DM, CP, GE, NDF, and EE in the HC group was the highest and significantly higher than that in the LC and MC groups (*p* < 0.05). There was no significant difference between LC and MC groups (*p >* 0.05). The digestibility of ADF in the HC group was significantly higher than that in the MC group, and there was no significant difference between the LC and the other groups. (*p >* 0.05).

### 3.2. Blood Parameters

The blood parameters of HPV are shown in Table 3. The concentrations of LA and VFAs in the HPV of the three groups linearly increased with the increase in dietary concentrate proportion, and the concentrations of the HC group were significantly higher than those of the LC and MC groups (*p* < 0.05). The concentration of LPS in the HC group was significantly higher than that of the LC and MC groups (*p* < 0.05). The differences in TG and NEFA among groups were not significant (*p* > 0.05). The LC group had a significantly higher GLU concentration compared to the MC and HC groups (*p* < 0.05).

The blood parameters of the JV are shown in Table 4. There was no significant difference in the concentrations of GLU, VFAs, ALT, BUN, TG, TP, ALB, LDL-C, HDL-C, LPS, LA, IL-6, TNF-α, IL-4, and LBP among groups (*p* > 0.05). The concentrations of IL-1β, DAO, and HP in the HC group were significantly higher than those in the LC and MC groups (*p* < 0.05), and the difference between the LC and MC groups was not significant (*p* > 0.05). The concentrations of IL-10 and SAA in the LC group were significantly lower than those in the MC and HC groups (*p* < 0.05).

### 3.3. Pathological Damage of Gastrointestinal Tract and Liver Tissues

The scores of tissue pathological damage are shown in Table 5. The slices showed that the liver injury in the MC group was slightly more serious than that in the LC and HC groups, but it did not reach a significant level among the groups (*p* > 0.05). In the LC and MC groups, the jejunal epithelial structure was intact, and a small number of inflammatory cells was found in the lamina propria and mucosa, whereas the degree of injury in the HC group was significantly higher than that in the LC and MC groups (*p* < 0.05), and there was a large amount of hemorrhage in the lamina propria, with more inflammatory cell infiltration. Although the colonic sections of the three groups revealed a little mucosal epithelium shedding, hyperemia, and lamina propria damage, there was no statistically significant difference among the groups (*p* > 0.05). The structure of the granular layer, spinous process layer, and basal layer of the rumen epithelium in the three groups were all normal and the tissue structure of all eighteen samples was intact and no obvious histopathological damage was recorded (Appendix A–D).

### 3.4. Transcriptome Analysis of Goat Liver

#### 3.4.1. Summary of RNA and Sequencing Quality Control

The present study established 18 cDNA libraries from the liver tissue of goats in the three groups and a total of more than six GO raw sequences were generated. After the removal of ribosomal RNA sequences and the filtering of low-quality sequences, the average clean sequences of the LC, MC, and HC groups were 406.92 ± 17.52 M, 420.73 ± 20.86 M, and 436.05 ± 33.63 M, respectively. The clean read rate of all samples was higher than 92.8%. The Q30 value of each library exceeded 92.58%. After mapping to the NCBI goat genome, 95.71% to 97.30% of clean sequences were successfully annotated (Appendix A).

#### 3.4.2. Differentially Expressed Genes among Groups

In the present RNA-seq study, 51,817 genes were detected in the liver of all 18 individuals. A total of 1119 DEGs were identified, among which 333 and 608 genes were up- and down-regulated in the HC group compared with the LC group, 80 and 19 genes were up- and down-regulated in the LC group compared with the MC group, and 89 and 140 genes were up- and down-regulated in the MC group compared with the HC group, respectively (Appendix A).

#### 3.4.3. Analysis of GO Functional Enrichment of DEGs

To explore the biological functions of the DEGs among groups, GO (gene ontology) enrichment analysis was performed. The DEGs of the LC vs. MC group, LC vs. HC group, and HC vs. MC group were enriched by 286 GO terms, 718 GO terms, and 410 GO terms (*p* < 0.05), respectively (Figure 1). These DEGs were classified into three main categories: biological process (BP), cellular component (CC), and molecular function (MF). Figure 2 showed the top ten enriched GO terms of the three main functional categories (BP, CC, and MF) in a pairwise comparison between the two groups. At the CC, MF, and BP levels, the most enriched GO terms for DEGs were endomembrane system, cation binding, and cholesterol metabolic process between the LC and HC groups (Figure 2A); respiratory chain complex, hydrogen ion transmembrane transporter activity, and electron transport chain between the LC and MC groups (Figure 2B); and chylomicron, gap junction channel activity, and atrial ventricular junction remodeling between the MC and HC groups (Figure 2C).

#### 3.4.4. Analysis of the KEGG Functional Enrichment of DEGs

The results of the KEGG enrichment analysis of DEGs are shown in Figure 3. In the pairwise comparison between the LC and MC groups and MC and HC groups, DEGs were enriched in 68 and 197 KEGG pathways, respectively. The DEGs of the LC vs. HC group enriched the largest number of KEGG pathways (289 in total), including 21 for cellular processes, 30 for environmental information processing, 19 for genetic information processing, 78 for human diseases, 62 for metabolism, and 79 for organismal systems. Figure 4 shows the top 20 most significantly enriched KEGG pathways of DEGs in the pairwise comparison between two groups. Four of the top 20 KEGG pathways of the LC vs. HC group were enriched in organismal systems such as the NOD-like receptor signaling pathway and the IL-17 signaling pathway and three were enriched in environmental information processing such as the MAPK signaling pathway and TNF signaling pathway (Figure 4A). Ten of the top twenty KEGG pathways for the LC vs. MC group were enriched for organismal systems such as thermogenesis and collecting duct acid secretion (Figure 4B). Eight of the top 20 Pathways of the MC vs. HC group were enriched for organismal systems such as cholesterol metabolism and the AMPK signaling pathway (Figure 4C).

#### 3.4.5. DEGs Related to Nutrient Metabolism and Toxin Detoxification

The DEGs of the LC vs. HC group, LC vs. MC group, and MC vs. HC group enriched a number of KEGG pathways, including cellular processes, genetic information processing, human diseases, and so on. Combining the results of the GO enrichment analysis and the purpose of this experiment, signaling pathways involved in nutrient metabolism and toxin metabolism detoxification were selected for demonstration, including AMPK, PPAR, insulin, thermogenesis, NF-κB, and toll-like receptor signaling pathways (Table 6). For these 6 pathways, the LC vs. HC group had the largest number of DEGs. Among these DEGs, the CCNA1, CREB3L4, and CPT1 genes were involved in the AMPK signaling pathway and were up-regulated in the LC group, whereas the CREB3L1, PFKFB1, LOC108637886, PRKAB1, HMGCR, MAP3K7, ACACA, and SIRT1 genes were down-regulated. In the thermogenesis signaling pathway, 11 genes (ATP6, COX7A1, CREB3L4, SLC25A29, ND6, CPT1B, COX2, LOC102173562, ATP5I, NDUFB9, and LOC102168533) were down-regulated in the LC group, whereas 6 genes (CREB3L1, KLB, SOS1, PRKAB1, SIRT6, and PRKACA) were up-regulated. Ten DEGs were involved in the toll-like receptor signaling pathway, two (IRF7 and TRAF3) of which were up-regulated in the LC group, whereas eight genes (TLR7, TLR4, CHUK, TAB2, MAP3K7, MAPK8, FADD, and IFNAR2) were down-regulated.

### 3.5. Validation of RNA-Seq Results by qRT-PCR

Eight DEGs (HMGCR, EIF4EBP1, ACCS, OXCT1, APOA1, APOC3, APOA5, and AACS) were quantified using real-time qPCR. The results (Figure 5) showed that in the HC group, the expression of APOA1, APOA5, and APOC3 was significantly higher, whereas the expression of ACCS, OXCT1, EIF4EBP1, and AACS was significantly lower than those in the LC and MC groups (*p* < 0.05). These results were the same in the transcriptomic analyses. The expression of HMGCR decreased linearly with the increase in dietary concentrate level and the difference among the three dietary groups was significant (*p* < 0.05), which was different from the results of the transcriptomic analysis. These results showed that the expression profiles of these genes detected by qRT-PCR were significantly similar to those detected by transcriptome, which confirmed the reliability of our RNA sequencing data.

## 4. Discussion

The present trial deals with the effects of high-concentrate diets on blood, rumen fermentation, and intestinal histomorphology in goats and explores the functional response of goat liver to HCD using transcriptome sequencing techniques.

Feeding high-concentrate diets to lactating cows results in the release of bacterial endotoxins, such as lipopolysaccharide (LPS), from the rumen or hindgut [24]. Free LPS can translocate into the bloodstream from the digestive tract under conditions of high permeability and after injury to the epithelial tissue [25]. LPS is a component of the cell wall of Gram-negative bacteria [5]. When the ruminal pH value significantly reduces, Gram-negative bacteria die and disintegrate and a large amount of LPS is released into the rumen fluid [26]. Research has shown that an accumulation of LPS in the rumen can damage the epithelial function and lead to pathological damage. When LPS enters the intestine with rumen chyme, it causes damage to the structure of the intestine [27]. When LPS is absorbed into the body, it triggers an immune response in the liver that releases inflammatory factors, which can cause further damage to the intestinal mucosa. Although the pathological section of the rumen epithelial tissue structure showed no significant difference in the degree of injury among groups, it was found that the jejunum tissue structure in the HC group had the most serious injury, with more inflammatory cell infiltration. In addition, the concentration of DAO in the JV blood of the HC group was also significantly higher than that of the other groups, indicating that the intestinal inflammatory injury in the HC group was the heaviest.

The liver is the most important metabolism and detoxification organ. Substances absorbed by the gastrointestinal tract, including nutrients and toxins, all enter the liver via HPV for decomposition, synthesis, and detoxification. The liver can control the processes involved in various ways, including altering the gene expression, protein synthesis, metabolite concentrations, enzyme activity, and nutrient fluxes produced, which can be optimized, and nutrients interconverted through liver function [28,29]. In this study, it was found that the concentration of VFAs in the HPV blood of the HC group was significantly higher than those of the other groups, indicating that the amount of acetic acid, propionic acid, and butyric acid absorbed by the HC group was higher than the other groups, whereas there was no significant difference in the JV blood among groups, which indicated that VFAs had been effectively metabolized in the liver. The acetic acid that enters the liver undergoes oxidation in the tricarboxylic acid cycle to produce energy or be catalyzed by acetyl-CoA carboxylase (ACC) to generate malonyl-CoA and then synthesize long-chain fatty acids [30].

Propionic acid entering the liver either generates glucose through gluconeogenesis or enters the tricarboxylic acid cycle to be oxidized to provide energy. The results of liver transcriptome sequencing in this study showed that the expression of *ACACA*, the gene encoding ACC, was significantly up-regulated in the HC group compared with the LC and MC groups in the AMPK signaling pathway and insulin signaling pathway. ACC is the rate-limiting enzyme of fatty acid biosynthesis, and the up-regulated expression of *ACACA* suggests that the biosynthesis of long-chain fatty acids from acetic acid was increased. It was also found that in the thermogenesis signaling pathway, the genes *ATP6*, *COX7A1*, *CPT1B*, *COX2*, *ATP5I*, and *NDUFB9* were significantly up-regulated in the HC group. These genes encode the key enzymes of the oxidative phosphorylation process [31], and their up-regulated expression indicated that the oxidation of VFAs to provide energy was also enhanced.

The calculation based on dietary chemical composition and nutrient digestibility showed that the goats in the HC group absorbed significantly more nutrients than goats in the LC group. For example, the daily absorption of EE in the HC group was 44.99 g/d, which was much higher than that in the LC group (21.12 g/d). A substantial increase in nutrient intake did not lead to hyperlipidemia and hyperglycemia in goats (the concentrations of GLU, TG, LDL-C, and HDL-C in the JV blood were not significantly different among groups). This was attributed to the enhanced metabolic function of the liver. In the AMPK signaling pathway of the present study, the up-regulated expression of genes involved in glucose (*PFKFB1*), lipid (*HMGCR*), and energy metabolism (*PRKAB1*) in the HC group suggests increased nutrient absorption and enhanced metabolic function. The increased absorption of nutrients also up-regulated the expression of many genes in the insulin signaling pathway. The up-regulation of *SOCS3* and *SOCS2* in the HC group suggests a potential role in regulating growth rate [32].

In ruminants, the liver is the major site for gluconeogenesis and lipogenesis [33]. A previous study showed that the concentrations of LPS in the rumen and the HPV blood of dairy goats consuming HCD were significantly increased [34], which was consistent with the results of this study. LPS entering the body is mainly inactivated and cleared in the liver [35,36]. In this experiment, although the concentration of LPS in the HPV blood of the HC group was 121.32% higher than that of the LC group, there was no significant difference between the two groups in the JV blood, which was consistent with the results of a previous study acting on cattle [37]. These results indicated that under the current experimental conditions, the liver of goats in the HC group had the ability to remove and detoxify excessive LPS entering the body. The liver detoxifies LPS mainly by its innate immune cells, which include Kupffer cells (KCs), dendritic cells (DCs), etc. [38]. As a major site for synthesizing circulating inflammatory cytokines [39,40], the liver could control the homeostasis balance by responding to both endogenous and exogenous stimuli [33]. When the liver is stimulated by LPS, it up-regulates the gene expression of innate immune cells and enhances the ability of the innate immune cells to recognize LPS, thereby eliminating LPS. In this experiment, the gene encoding TLR4 and TLR7 proteins in the HC group was significantly up-regulated, and TLR proteins are key molecules in the LPS signal transduction pathway. Toll-like receptors (TLRs) play a key role in maintaining homeostasis by recognizing ligands known as microbial-associated molecular patterns derived from both pathogenic and nonpathogenic bacteria [41].

The up-regulated expression of TLR genes can enhance the recognition of LPS by liver immune cells such as KCs and present LPS to phagocytes so that it can be quickly cleared by the liver. Meanwhile, in this experiment, the genes encoding apolipoproteins (APO-A1, APO-A2, APO-A5, and APO-C3) in the PPAR signaling pathway were also specifically up-regulated in the HC group. Apolipoprotein APO-A1 is a major component of high-density lipoprotein (HDL) [42], which has been shown to neutralize the biological activity of LPS [43]. Although the liver of the HC group in this experiment can effectively eliminate LPS by up-regulating the expression of genes such as *TLR4* and apolipoprotein *APO-A1*, the KCs and other immune cells produced a large number of inflammatory factors during the process of activating NF-κB and clearing LPS.

Although the liver of the HC group in this experiment can effectively eliminate LPS by up-regulating the expression of genes such as TLRs and apolipoproteins, the KCs and other immune cells produced a large amount of pro-inflammatory cytokines during the process of activating the NF-κB signaling pathway and clearing LPS. This was the reason why the concentrations of IL-1β, IL-6, and TNF-α in the JV blood in the HC group were significantly higher than those in the LC and MC groups. While pro-inflammatory cytokines can aid in LPS clearance, excessive amounts can lead to liver cell apoptosis and damage. In this experiment, it was found that the up-regulated expression of *FADD* in the HC group suggests a potential role in inducing cell apoptosis in response to toll-like receptor signaling, which was consistent with the results of a previous trial [44]. These results indicate that goats can be suitable for HCDs in the short term, but the long-term feeding of HCDs may induce extensive inflammation, reduce immunity, and ultimately damage the health of goats.

## 5. Conclusions

Feeding goats with an HCD can significantly increase the absorption of potentially harmful substances such as LPS and toxins. The liver of goats can adapt to an HCD by increasing the metabolism of nutrients and the detoxification of harmful substances through the regulation of key genes. This study revealed the mechanism of HCD application in goats from the level of gene expression and laid a foundation for the rational application of HCDs in goats. It is worth noting that this study was officially tested for only 67 days. Although the goats developed adaptability to HCDs during this short feeding period, the long-term feeding of an HCD may increase the risk of chronic inflammation and other health problems in goats.

## Figures and Tables

**Figure 1 animals-13-01559-f001:**
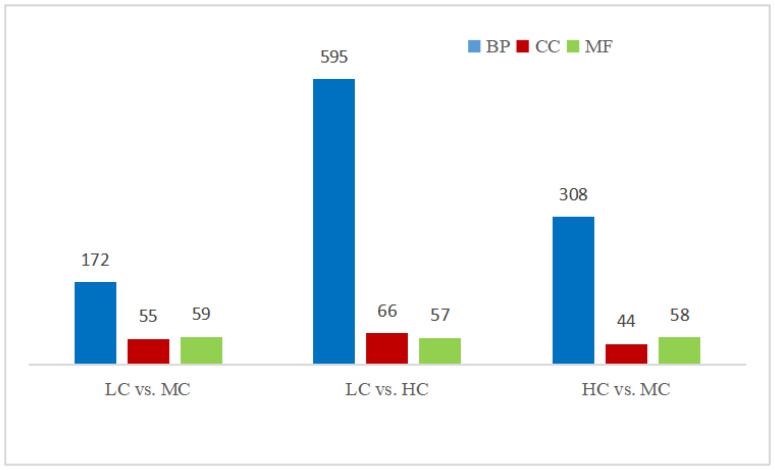
Number of different gene ontology (GO) terms for differentially expressed genes (DEGs) in goats between groups. HC = high level of concentrate diet group (n = 6); MC = medium level of concentrate diet group (n = 6); LC = low level of concentrate diet group (n = 6). BP = biological process; CC = cellular component; MF = molecular function.

**Figure 2 animals-13-01559-f002:**
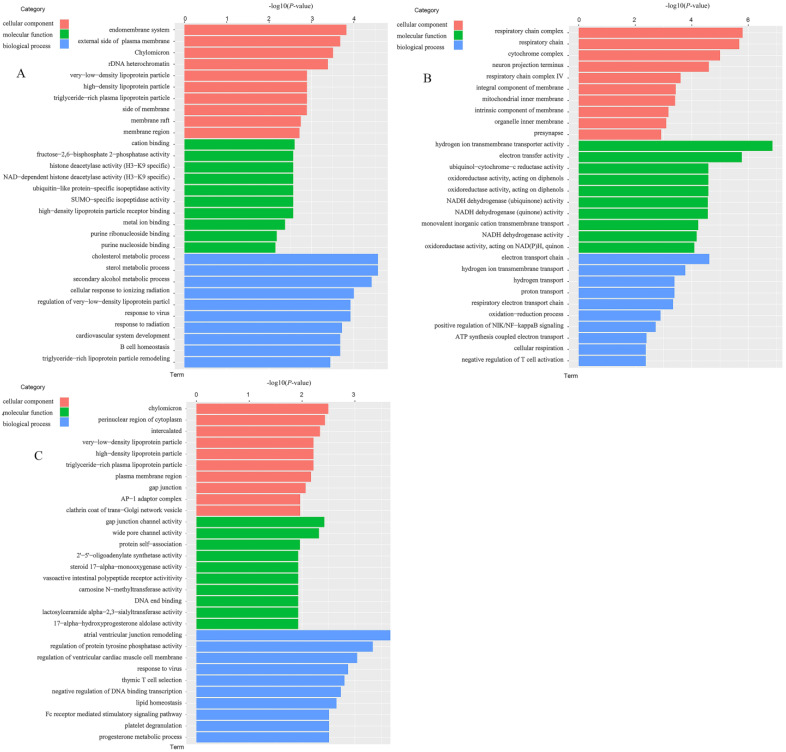
Top 10 gene ontology (GO) terms of the GO enrichment analysis of differentially expressed genes between groups. (**A**) Between LC and HC groups; (**B**) between LC and MC groups; (**C**) between MC and HC groups. HC = high level of concentrate diet group (n = 6); MC = medium level of concentrate diet group (n = 6); LC = low level of concentrate diet group (n = 6).

**Figure 3 animals-13-01559-f003:**
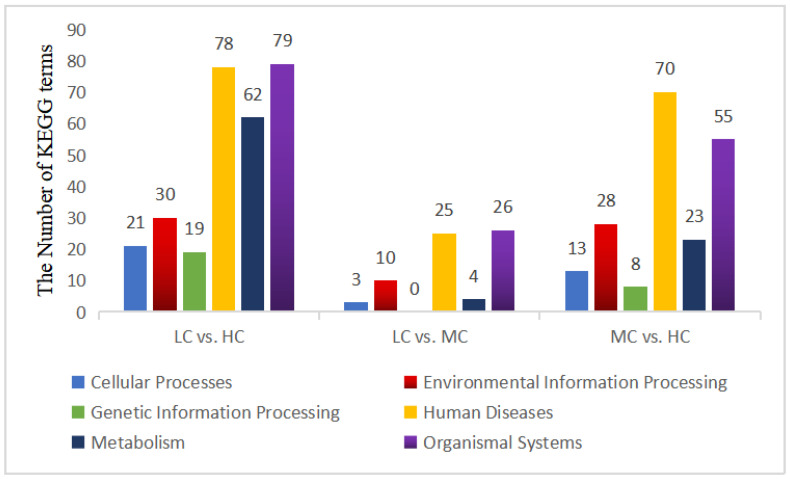
KEGG enrichment analysis of differentially expressed genes. HC = high level of concentrate diet group (n = 6); MC = medium level of concentrate diet group (n = 6); LC = low level of concentrate diet group (n = 6).

**Figure 4 animals-13-01559-f004:**
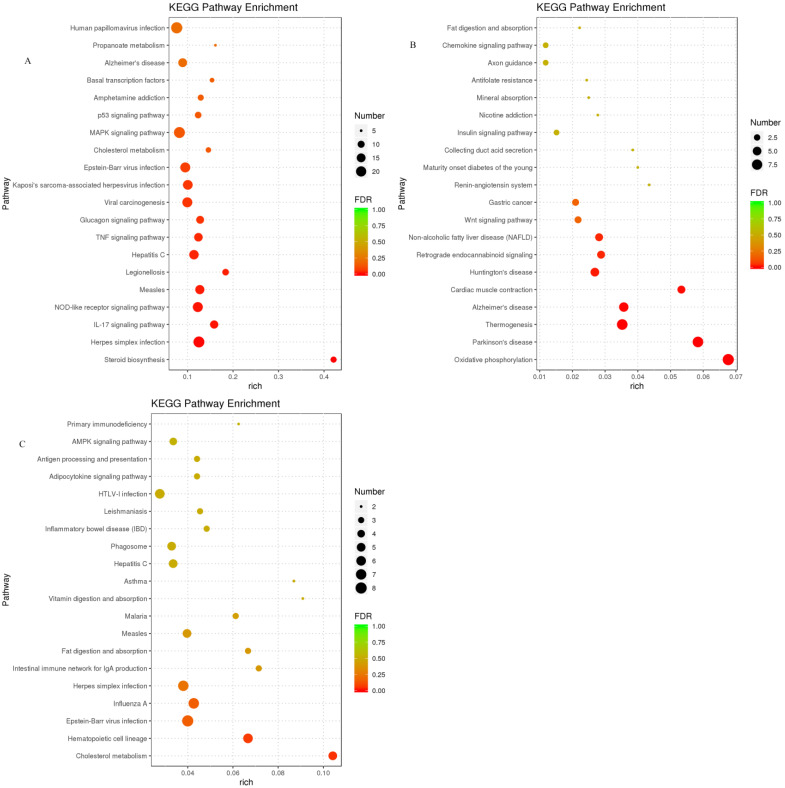
Top 20 pathways of the KEGG enrichment analysis between groups. The bubble size represents the number of differentially expressed genes (DEGs) enriched in the pathway, and the bubble color represents the false discovery rate value. (**A**) Between LC and HC groups; (**B**) between LC and MC groups; (**C**) between MC and HC groups. HC = high level of concentrate diet group (n = 6); MC = medium level of concentrate diet group (n = 6); LC = low level of concentrate diet group (n = 6).

**Figure 5 animals-13-01559-f005:**
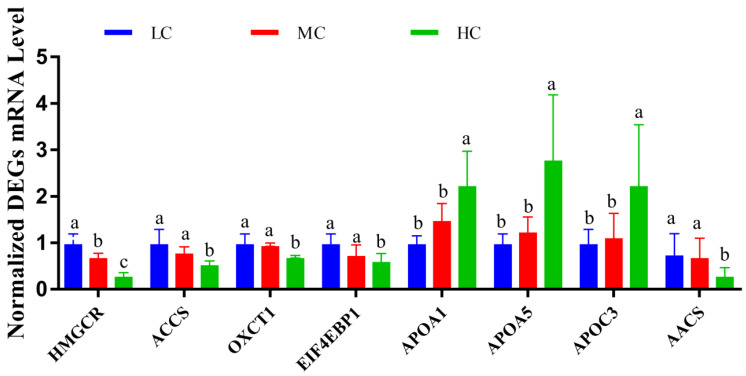
Quantitative real-time PCR validation of the selected differentially expressed genes (DEGs). a, b, c Means with different superscripts within the same column differ significantly (*p* < 0.05). Values are presented as means with their standard errors. The relative quantification of the DEGs was calculated with the formula 2^−ΔΔCt^ and normalized to β-actin control. HC = high level of concentrate diet group (n = 6); MC = medium level of concentrate diet group (n = 6); LC = low level of concentrate diet group (n = 6).

**Table 1 animals-13-01559-t001:** Ingredients and chemical composition of experimental diets (%, dry matter basis).

Item	Group ^2^		
	LC	MC	HC
Ingredients			
Corn	4.00	28.67	49.25
Wheat bran	7.00	2.71	7.30
Soybean meal	9.00	6.78	10.50
Rapeseed meal	1.00	6.32	1.00
Cottonseed meal	1.50	2.26	8.00
Oat grass	43.80	24.00	5.00
Wheat straw	8.00	6.00	1.00
Alfalfa hay	23.00	20.00	14.00
CaCO_3_	0.15	0.50	1.00
CaHPO_4_	0.05	0.26	0.45
NaCl	0.50	0.50	0.50
NaHCO_3_	1.00	1.00	1.00
Premix ^1^	1.00	1.00	1.00
Total	100.00	100.00	100.00
Chemical composition			
Dry matter	88.84	89.05	90.14
Crude protein	12.06	13.77	15.93
Ether extract	3.08	4.23	5.23
Crude fiber	23.19	16.42	8.71
Nitrogen free extract	35.24	42.11	50.89
Metabolizable energy, MJ/kg	2.17	2.49	2.88
Neutral detergent fiber	43.54	31.65	19.55
Acid detergent fiber	21.05	15.04	7.34
Calcium	0.59	0.67	0.76
Phosphorus	0.37	0.43	0.49
Concentrate:roughage	25:75	50:50	80:20

^1^ The premix provided the following per kg of diet: vitamin A, 7000 IU; vitamin D, 500 IU; vitamin E, 50 IU; Fe, 40 mg; Cu, 10 mg; Zn, 30 mg; Mn, 40 mg; I, 0.8 mg; Se, 0.2 mg; Co, 0.11 mg. ^2^ HC = high level of concentrate diet group; MC = medium level of concentrate diet group; LC = low level of concentrate diet group.

**Table 2 animals-13-01559-t002:** Apparent total tract digestibility of nutrients (%) of different groups in goats.

Item	Group ^1^		
	LC	MC	HC
Dry matter	64.30 ± 2.99 ^b^	62.70 ± 4.84 ^b^	79.44 ± 3.59 ^a^
Crude protein	70.30 ± 5.94 ^b^	67.10 ± 5.56 ^b^	80.48 ± 3.80 ^a^
Gross energy	63.24 ± 2.94 ^b^	61.45 ± 4.69 ^b^	79.08 ± 3.54 ^a^
Acid detergent fiber	49.43 ± 5.19 ^ab^	34.74 ± 9.36 ^b^	51.94 ± 8.84 ^a^
Neutral detergent fiber	54.86 ± 6.15 ^b^	47.68 ± 8.72 ^b^	64.24 ± 6.12 ^a^
Ether extract	59.59 ± 3.60 ^b^	58.73 ± 4.82 ^b^	79.65 ± 3.33 ^a^

^a, b^ Means with different superscripts within the same column differ significantly (*p* < 0.05). ^1^ HC = high level of concentrate diet group; MC = medium level of concentrate diet group; LC = low level of concentrate diet group.

**Table 3 animals-13-01559-t003:** Hepatic portal vein blood parameters of different groups in goats.

Item	Group ^1^		
	LC	MC	HC
Glucose, g/L	6.70 ± 1.66 ^a^	5.04 ± 0.50 ^b^	5.36 ± 0.55 ^b^
Triglyceride, g/L	0.69 ± 0.08	0.69 ± 0.08	0.67 ± 0.09
Non-esterified fatty acid, mmol/L	0.11 ± 0.03	0.09 ± 0.03	0.12 ± 0.01
Lipopolysaccharide, EU/L	203.65 ± 31.76 ^b^	193.43 ± 10.41 ^b^	247.07 ± 27.70 ^a^
Lactic acid, μg/L	496.67 ± 422.93 ^b^	841.25 ± 668.99 ^b^	2093.75 ± 759.75 ^a^
Acetate, μmol/L	701.66 ± 146.24 ^b^	918.13 ± 119.80 ^b^	1227.25 ± 313.51 ^a^
Propionic, μmol/L	72.95 ± 10.54 ^b^	95.92 ± 10.55 ^b^	114.33 ± 12.52 ^a^
Butyrate, μmol/L	46.37 ± 6.21 ^b^	52.38 ± 6.58 ^b^	65.68 ± 15.53 ^a^

^a, b^ Means with different superscripts within the same column differ significantly (*p* < 0.05). ^1^ HC = high level of concentrate diet group; MC = medium level of concentrate diet group; LC = low level of concentrate diet group.

**Table 4 animals-13-01559-t004:** Jugular vein blood parameters of different groups in goats.

Item	Group ^1^		
	LC	MC	HC
Glucose, g/L	3.33 ± 0.75	3.08 ± 0.97	3.73 ± 1.08
Acetate, μmol/L	60.68 ± 18.50	46.77 ± 16.72	47.91 ± 10.42
Propionic, μmol/L	10.73 ± 2.12	9.20 ± 1.25	9.36 ± 2.45
Butyrate, μmol/L	3.91 ± 2.82	5.13 ± 1.91	6.09 ± 1.21
Alanine transaminase, g/L	24.33 ± 10.07	18.17 ± 5.49	23.50 ± 7.15
Blood urea nitrogen, g/L	7.32 ± 3.08	6.60 ± 1.40	8.58 ± 1.87
Triglyceride, g/L	0.57 ± 0.09	0.51 ± 0.06	0.50 ± 0.04
Total protein, g/L	70.92 ± 6.22	73.48 ± 9.31	65.37 ± 5.58
Albumin, g/L	34.32 ± 2.56	35.52 ± 3.50	34.57 ± 1.56
Low-density lipoprotein-C, mmol/L	0.34 ± 0.13	0.25 ± 0.10	0.25 ± 0.10
High-density lipoprotein-C, mmol/L	1.04 ± 0.21	1.21 ± 0.34	1.16 ± 0.30
Lipopolysaccharide, EU/L	202.71 ± 10.02	189.01 ± 13.09	201.09 ± 21.91
Lactic acid, μg/L	2.94 ± 0.67	4.38 ± 0.21	2.67 ± 0.12
Interleukin-1β, ng/L	234.44 ± 24.94 ^b^	247.71 ± 28.03 ^b^	288.29 ± 62.02 ^a^
Interleukin-6, ng/L	104.72 ± 40.93	128.19 ± 40.18	115.64 ± 19.25
Tumor Necrosis Factor-α, ng/L	1757.60 ± 230.23	1758.60 ± 531.31	1796.74 ± 116.66
Interleukin-4, ng/L	94.15 ± 23.25	127.74 ± 69.16	155.48 ± 36.20
Interleukin-10, ng/L	1279.65 ± 186.63 ^b^	2217.24 ± 767.27 ^a^	2319.94 ± 423.92 ^a^
Diamineoxidase, pg/ml	204.89 ± 18.52 ^b^	210.38 ± 15.88 ^b^	273.99 ± 11.94 ^a^
Lipopolysaccharide-binding protein, pg/mL	10.07 ± 0.42	9.63 ± 0.47	10.09 ± 0.51
Haptoglobin, pg/mL	14.60 ± 1.04 ^b^	16.93 ± 0.60 ^b^	18.53 ± 0.85 ^a^
Serum amyloid A protein, pg/mL	439.20 ± 13.46 ^b^	530.35 ± 31.67 ^a^	548.79 ± 22.35 ^a^

^a, b^ Means with different superscripts within the same column differ significantly (*p* < 0.05). ^1^ HC = high level of concentrate diet group; MC = medium level of concentrate diet group; LC = low level of concentrate diet group.

**Table 5 animals-13-01559-t005:** The tissue pathological damage score of different groups in goats.

Tissue ^2^	Group ^1^		
	LC	MC	HC
Liver	1.33 ± 0.52	1.83 ± 0.75	1.50 ± 0.84
Jejunum	1.00 ± 0.63 ^b^	0.83 ± 0.75 ^b^	2.17 ± 0.98 ^a^
Colon	1.80 ± 0.45	2.00 ± 0.00	1.80 ± 0.45
Rumen	0.00 ± 0.00	0.00 ± 0.00	0.00 ± 0.00

^a, b^ Means with different superscripts within the same column differ significantly (*p* < 0.05). ^1^ HC = high level of concentrate diet group; MC = medium level of concentrate diet group; LC = low level of concentrate diet group. ^2^ Tissue pathological damage of the rumen, liver, jejunum, and colon was scored from 1 to 3 referring to the previously described methods [19,20,21], respectively, with a score of 0 denoting no damage and a score of 3 denoting severe damage.

**Table 6 animals-13-01559-t006:** Significantly enriched KEGG pathways of down-regulated and up-regulated genes.

Pathway ID	Pathway Name	Groups ^1^					
		LC vs. HC		LC vs. MC		MC vs. HC	
		Up-Regulated Gene	Down-Regulated Gene	Up-Regulated Gene	Down-Regulated Gene	Up-Regulated Gene	Down-Regulated Gene
chx04152	AMPK signaling pathway	CCNA1, CREB3L4, CPT1	CREB3L1, PFKFB1, LOC108637886, PRKAB1, HMGCR, MAP3K7, SIRT1, ACACA			CCNA1	PFKFB1, LOC108633303, ACACA
chx03320	PPAR signaling pathway	EHHADH	APOA5, CPT1B, APOA1, APOA2, APOC3	SORBS1		LOC108633303	APOA1, APOC3
chx04910	Insulin signaling pathway	PRKAR1B, CALML4	SOCS3, SOCS2, SOS1, PRKAB1, ACACA, MAPK8, PPP1R3C, PPP1CB, EIF4E, PRKACA	SHC2, SORBS1			SOCS3, ACACA, SHC2, EIF4E
chx04714	Thermogenesis signaling pathways	CREB3L1, KLB, SOS1, PRKAB1, SIRT6, PRKACA	ATP6, COX7A1, CREB3L4, SLC25A29, ND6, CPT1B, COX2, LOC102173562, ATP5I, NDUFB9, LOC102168533	ND1, ATP6, ND5, LOC102168533, CYTB, ND6, COX3, COX2		COX7A1	
chx04064	NF-κB signaling pathway	TRAF3	CHUK, LYN, TAB2, MAP3K7, NFRSF13C, ICAM1, IL1R1				TLR4, LYN, TRAF5
chx04620	Toll-like receptor signaling pathway	IRF7, TRAF3	TLR4, TLR7, CHUK, TAB2, MAP3K7, MAPK8, IFNAR2, FADD		TLR7	IRF7	TLR4

^1^ HC = high level of concentrate diet group; MC = medium level of concentrate diet group; LC = low level of concentrate diet group.

## Data Availability

Original data are available from the corresponding author upon reasonable request. The RNA-seq data are available in the NCBI under BioProject PRJNA385353.

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
