# Peer review of "Effects of a High-Concentrate Diet on the Blood Parameters and Liver Transcriptome of Goats"

_animals, 2023, doi:10.3390/ani13091559_

Round 1

Reviewer 1 Report

The manuscript prepared by the authors makes a useful contribution to the literature. Most of the reviewer's comments are recorded in comments boxes associated with the PDF document attached to this review.

The paper focuses on goats from a particular region in China. For this reason, it is important to contextualize the geographic locations for sheep and goat production that the authors refer to in the introduction. Some of the statements made in the introduction are poorly referenced. The published caprine literature in the field should be mentioned and cited.

Some details are lacking under experimental animals, design, and diet. These are important for others to reproduce the scientific method reported by the authors.

The text associated with some of the figures is so poor that it was not possible for the reviewer to read them. This will need to be rectified before publication.

The discussion is concise and generally applicable.  However, the authors have not cited several caprine references which would better support their discussion. Whereas it is helpful to compare and contrast with another species, where possible key comparisons should be made with the published literature for the goat. Examples are provided below:

Dong H, Wang S, Jia Y, Ni Y, Zhang Y, Zhuang S, et al. (2013) Long-Term Effects of Subacute Ruminal Acidosis (SARA) on Milk Quality and Hepatic Gene Expression in Lactating Goats Fed a High-Concentrate Diet. PLoS ONE 8(12): e82850. https://doi.org/10.1371/journal.pone.0082850

Duanmu Y, Cong R, Tao S, Tian J, Dong H, Zhang Y, Ni Y, Zhao R. Comparative proteomic analysis of the effects of high-concentrate diet on the hepatic metabolism and inflammatory response in lactating dairy goats. J Anim Sci Biotechnol. 2016 Feb 6;7:5. doi: 10.1186/s40104-016-0065-0

Tao S, Duanmu Y, Dong H, Tian J, Ni Y, Zhao R. A high-concentrate diet induced colonic epithelial barrier disruption is associated with the activating of cell apoptosis in lactating goats. BMC Vet Res. 2014 Sep 26;10:235. doi: 10.1186/s12917-014-0235-2

Ianni A, Bennato F, Martino C, Colapietro M, Martino G. Whole Blood Transcriptome Profiling Reveals Positive Effects of Olive Leaves-Supplemented Diet on Cholesterol in Goats. Animals (Basel). 2021 Apr 17;11(4):1150. doi: 10.3390/ani11041150. PMID: 33920539; PMCID: PMC8072609.

 The conclusions drawn by the authors are appropriate.

Author Response

We greatly appreciate the insightful and careful review by the reviewer and editors, and we have made the following major revisions:

We have checked our manuscript and corrected language errors carefully. We have also cited new literature as requested.

Please see the attachment for a point-by-point responses to the reviewer.

Reviewer 2 Report

Dear authors, please see my comments in the attached paper, best

Author Response

We greatly appreciate the insightful and careful review by the reviewer and editors, and we have made the following major revisions:

We have checked our manuscript and corrected language errors carefully. 

Please see the attachment for a point-by-point responses to the reviewer.

Round 2

Reviewer 2 Report

The paper has improved a lot. Good job!